# Abrupt light transitions in illuminance and correlated colour temperature result in different temporal dynamics and interindividual variability for sensation, comfort and alertness

**Maaike E. Kompier** [ID]*, **Karin C. H. J. Smolders, Yvonne A. W. de Kort**

Human-Technology Interaction, School of Innovation Sciences, Eindhoven University of Technology, Eindhoven, the Netherlands

* m.e.kompier@tue.nl

**Data Availability Statement:** Data cannot be shared publicly because of the informed consent form stating that data will be only available to

## Abstract

Detailed insights in both visual effects of light and effects beyond vision due to manipulations in illuminance and correlated color temperature (CCT) are needed to optimize study protocols as well as to design light scenarios for practical applications. This study investigated temporal dynamics and interindividual variability in subjective evaluations of sensation, comfort and mood as well as subjective and objective measures of alertness, arousal and thermoregulation following abrupt transitions in illuminance and CCT in a mild cold environment. The results revealed that effects could be uniquely attributed to changes in illuminance or CCT. No interaction effects of illuminance and CCT were found for any of these markers. Responses to the abrupt transitions in illuminance and CCT always occurred immediately and exclusively amongst the subjective measures. Most of these responses diminished over time within the 45-minute light manipulation. In this period, no responses were found for objective measures of vigilance, arousal or thermoregulation. Significant interindividual variability occurred only in the visual comfort evaluation in response to changes in the intensity of the light. The results indicate that the design of dynamic light scenarios aimed to enhance human alertness and vitality requires tailoring to the individual to create visually comfortable environments.

## Introduction

The light in our direct surroundings affects us in more ways than many of us are aware of, and has implications for what we see, how we behave and how we feel. Researchers studying responses to light today speak of two major pathways through which light acts on human physiology and psychology: a visual pathway and a non-visual, or non-image forming pathway [1]. The first pathway originates in the rods and cones in our eyes, which allow us to see the environment around us [2]. As a strongly visually oriented species, the focus of research and

others in an encrypted and password protected institutional online data repository. Data are available upon request via the OSF project page through which requests can be sent to all authors (https://osf.io/34jr7/) or via the Ethics Committee (contact via ethicalreviewboardHTI@tue.nl).

**Funding:** This research is part of the DYNKA project (TEUE117001) funded under the TKI Urban Energy scheme by the Top consortium for Knowledge and Innovation - Topsector Energy of the Dutch Ministry of Economic Affairs (www. topsectorenergie.nl/). The funders had no role in study design, data collection and analysis, decision to publish, or preparation of the manuscript.

**Competing interests:** The authors have declared that no competing interests exist.

standards was for a long time directed towards the requirements for light to assure proper visual performance, experience and comfort [3,4]. In the past two decades, the intrinsically photosensitive retinal ganglion cells (ipRGC) have been identified and studied as the primary driver for the second pathway; the effects beyond vision [5,6]. Such effects include the alignment of our internal 24h rhythm to the external rhythm of the world, i.e. so-called circadian effects [6]. Other ipRGC-influenced light responses (IIL) acutely influence our health, well-being and productivity [5]. Importantly, the relevant photoreceptors and underlying sensory systems show marked differences in terms of temporal, spatial, spectral and intensity-related processes [6–11]. To create light scenarios that positively promote health and wellbeing in terms of both visual experiences and light effects beyond vision, we may require light that is dynamic in illuminance and correlated color temperature (CCT) [12]. Several studies have investigated effects of such dynamic scenarios and hint towards positive circadian and acute effects [13–15]. However, these studies all differed considerably in the outcome parameters that were investigated and the measurement protocols that were used. In addition, the timing of the measurements within the light scenario and the exposure duration per condition varied substantially, preventing the formulation of clear-cut conclusions [12]. Preceding structured investigations of the overall effects of a day-long dynamic light scenario, the effects of such scenarios' individual elements (e.g., the transitions vs. the static parts) should be studied. More specifically, it is important to investigate and be aware of the changes in subjective states, behavior and physiology that may develop over time following a transition in light conditions. In the current study, we aim to examine the temporal trajectories of alertness, arousal, comfort and mood–and individual differences therein–immediately following an abrupt transition in light concerning illuminance and/or CCT using repeated measurements.

Effects of light on, for instance, sleep, human physiology, alertness and cognitive performance have been described for various exposure durations and levels of both illuminance [16–18] and CCT [19–21]. Although the knowledge gained through these studies is very valuable, most studies do not provide detailed information about the short-term development over time of the various responses (e.g., in terms of comfort, alertness, arousal and mood) that a change in light induces. Knowledge on the time it takes for an effect to emerge (**onset**) and whether the effect persists throughout the entire light exposure (**persistence**) is indispensable in the design of study protocols as well as practical light applications. This knowledge can be gained by continuous or repeated measurements of the dependent variable of interest after the onset of the light exposure. Although some laboratory studies did study participants' responses to specific lighting conditions repeatedly on an hourly basis throughout the light exposure [22,23], more frequent repeated measurements within one hour after onset of the light manipulation are also needed to determine the detailed temporal trajectories of the outcome parameters. Several studies reported that effects of a light manipulation on indicators of subjective alertness, physiological arousal and performance measures, if they emerged, were immediate and persistent [24–28]. Others, however, reported delayed onset of performance effects [29,30] or self-reported alertness [31]. One study reported significant effects of CCT on one of the performance tasks only mid-way throughout one hour of exposure [32]. Another study [33] reported fast responses (within the first 5 minutes) which then stabilized throughout 50 minutes for some measures (e.g., pupil response, EEG, heart rate), whereas other measures showed continued gradual incline or decline (e.g., HRV, DPG). The before-mentioned studies focused mainly on IIL responses and studied either illuminance or CCT, whereas the visual effects of light and potential interactions between illuminance and CCT are also crucially important in optimizing study protocols as well as light scenarios for practical applications.

A recently published study examined temporal dynamics of visual experiences and light effects beyond vision–for example, sensation, comfort, mood, self-reported alertness and

physiology–after abrupt transitions in illuminance and CCT and reported markedly different trajectories for different indicators [28]. Unfortunately, in this study changes in illuminances were always confounded with changes in CCT. Increases in both illuminance and CCT enhance the stimulation of the intrinsic photosensitive retinal ganglion cells (ipRGC) that are traditionally held responsible for the acute, alerting effects of light [34,35], but their effects on visual appraisals may work in opposite directions. Despite its alerting potential, cool, bright light is–at times–experienced as unsatisfying and uncomfortable [36], which then may also result in compromised alertness and performance [37]. This demonstrates that the effects of illuminance and CCT may converge and hence strengthen each other for some measures, but diverge, potentially neutralizing each other for other measures. Studies investigating the effects of illuminance and CCT simultaneously yet independently are still quite scarce and generally do not focus on the temporal trajectories of the outcome variables in response to the manipulation of one or both of these parameters [36,38–41]. This stresses the importance to study the temporal dynamics of responses to both light parameters simultaneously and disentangle these two factors.

To complicate matters, the literature on effects of illuminance and CCT has repeatedly suggested that the responsiveness for light responses is susceptible to large interindividual differences. Sometimes the lack or subtlety of effects of light on subjective sleepiness/alertness during daytime is attributed to interindividual differences [25,26,42]. A certain light manipulation may be beneficial for some people, but detrimental for others, resulting in small and/or insignificant overall effects, whereas on an individual level the effects are of great practical relevance. Some studies have explicitly shown the existence of substantial interindividual variability in responsiveness for different measures (e.g., visual comfort and circadian phase shifting responses of light) [43–45]. Potential explanations for these interindividual differences in responsiveness to light could be the dependence on individual's prior light exposure and thus be more situational [46,47], but it could also have a genetic basis and reflect trait rather than state differences [48,49]. In this study, we take a first step in statistically examining the existence of such significant interindividual variability in responses to light.

In order to investigate the temporal dynamics of the independent and joint effects of illuminance and CCT, we studied the effects of abrupt step-wise increases in illuminance and CCT on participants' evaluations of the ambient environment and their functioning. In this study we additionally examined the extent to which interindividual variability affects the responses to transitions in CCT and/or illuminance of the light. The employed study design also allows for a partial replication of the study by Kompier, Smolders, van Marken Lichterbelt and de Kort [28] on response dynamics after an abrupt transition in lighting. Despite the lack of effects on physiological and behavioral measures in Kompier et al. [28], these were included here to allow for replication of these null results. Whereas [28] was performed in thermoneutral conditions, the current study was performed in a mild cold environment as there are indications that slight discomfort may be a prerequisite for cross-modal interactions between thermal and visual comfort [50]. The research question the current study sought to answer was to what extent an abrupt transition from dim to bright light, or from warm to cool light, influences subjective evaluations of comfort and mood as well as subjective and objective measures of alertness, arousal and thermoregulation over time. We hypothesized fast, but transient responses of the subjective parameters (e.g., comfort, alertness and mood) based on the results by Kompier et al. [28]. Moreover, we sought to investigate the extent to which systematic interindividual variability occurs in these light-induced effects. It was hypothesized that these would exist for alertness and visual comfort, but not for the other variables. Last, we aimed to study the extent to which transitions in the illuminance and correlated colour temperature

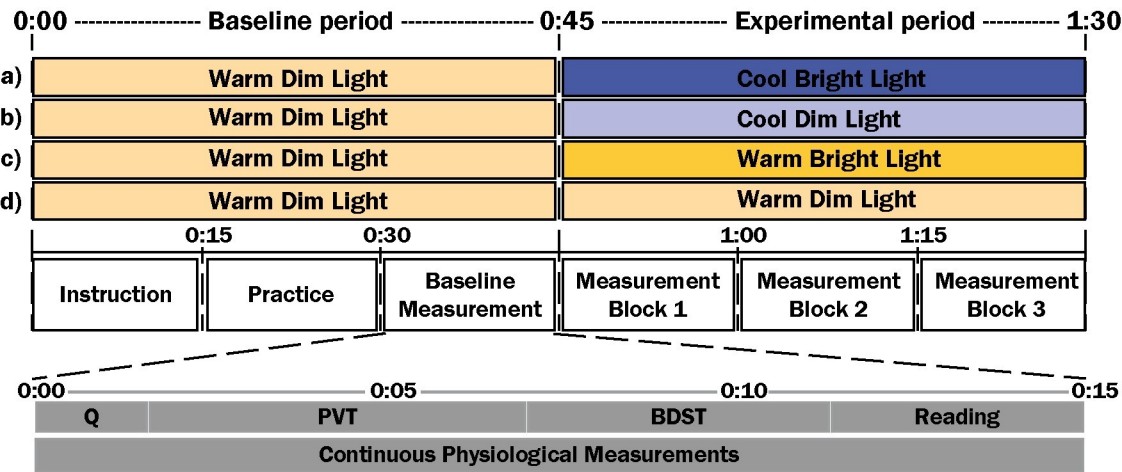

**Fig 1. The four light experimental light conditions and the experimental procedure of one session.** (A) cool, bright light (CB), (B) cool, dim light (CD), (C) warm, bright light (WB), and (D) warm, dim light (WD). Participants were exposed to these four conditions in separate sessions on different days in partial counterbalanced order. Q: Questionnaire, PVT: Psychomotor Vigilance Task, and BDST: Backward Digit Span Task.

(CCT) of the light interact. These interaction effects were expected to particularly prevail in the effect on visual comfort.

## Materials and methods

### Design

Temporal dynamics of responses to abrupt transitions in lighting were examined in a laboratory study with a two (Illuminance: bright vs. dim) by two (CCT: cool vs. warm) within-subjects design. Each session lasted 90 minutes, and existed of a baseline period of 45 minutes in which warm, dim light was administered. At the start of the 45-min experimental period following this baseline period, the light was changed to one of the four experimental light conditions (see also Fig 1). The experimental light conditions were a) Cool, bright light; b) Cool, dim light; c) Warm, bright light; and d) Warm, dim light. Participants were exposed to all of them in separate sessions on separate days. The order of the conditions in these sessions was counterbalanced. Self-report, performance and physiological markers were used as outcome measures. The study was approved by the institutional ethical review board (HTI Ethical Review Board—experiment ID 963). Participants gave their written informed consent and received a monetary compensation for their participation.

### Participants

Twenty-three healthy participants (13 female; $M_{age}$ = 23, $SD_{age}$ = 2.0; *range* = 18–26 years old) were recruited via the J.F. Schouten School for User-System Interaction Research database from the Eindhoven University of Technology. None of the participants reported visual or auditory deficits, or were an extreme chronotype (assessed using the Munich Chronotype Questionnaire [51]). Additionally, none of the participants used medication other than the contraceptive pill or suffered from hypertension or cardiovascular disease. Last, none of them travelled intercontinentally in the past three months. Further descriptors can be found in Table 1.

**Table 1. Participant descriptors.**

| | n = 23 | |
| --- | --- | --- |
| | Mean (± SD) | Range |
| Body Mass Index | 22 (± 3.0) | 18–30 |
| Midsleep (MSF$_{sc}$) | 4.7 (± 1.1) | 2.4–6.9 |
| PSQI score | 4.0 (± 2.4) | 1–10 |
| General Health items (SF-36) | 74 (± 14) | 45–100 |
| Light Sensitivity–Eye Problems | 1.8 (± 0.8) | 1–4 |
| Light Sensitivity–Headache | 1.7 (± 0.9) | 1–3 |
| Light Sensitivity–Sunglasses | 2.7 (± 1.3) | 1–5 |
| Thermal Sensitivity | 3.0 (± 1.1) | 1–4 |

## Setting and apparatus

An office setting with two separate desks was created in a climate chamber with a floor area of 18.0m$^2$ and a ceiling height of 2.7m. With a Minolta Luminance Meter LS-100, the reflectance of the various surfaces in the room were measured (off-white walls: 80.9%, white partition wall: 91.8%, grey floor: 27.7%, light grey desk: 49.0%). The intended air temperature in the room was 17.0˚C, but presumably due to the heat production of the participants the operative air temperature was 18.0˚C ± 0.1 (air velocity = 0.01m/s ± 0.00; relative humidity = 67.4% ± 7.2, and black bulb temperature = 17.6˚C ± 0.1). The four light settings were created using a set of four ceiling-mounted luminaires (PowerBalance Tunable Whites; RC464B LED80S/ TWH PSD W60L60) above each desk. Table 2 shows the alpha-opic equivalent daylight illuminances (EDI), illuminance and CCT of all four light settings, and Fig 2 shows the spectral power distribution of the four light settings.

## Measurements

**Subjective measurements.** Visual experience was probed as visual sensation, acceptance and comfort with items identical to the items in Kompier et al. [28]. Participants' visual sensation was measured with two separate items for perceived brightness and color (Sensation$_{VI}$ and Sensation$_{VC}$) on 7-point rating scales ranging from *Very Low (-3)* to *Very High (3)* and *Very Cool (-3)* to *Very Warm (3)* respectively. Furthermore, participants evaluated acceptance of the lighting (Acceptance$_V$) on a binary scale (*Acceptable/Unacceptable*). Participants were also asked to assess comfort with the brightness and color of the white lighting (Comfort$_{VI}$ and Comfort$_{VC,}$ respectively) on 6-point rating scales ranging from *Very Uncomfortable (-2)* to *Just Uncomfortable (0)* and from *Just Comfortable (1)* to *Very Comfortable (3)*. These two scores were averaged into Comfort$_V$ as they had an internal consistency (Cronbach's α) of 0.81.

**Table 2. α-opic EDI, illuminance and CCT on the eye of the light settings.**

| | Cool, bright light (CB) | Cool, dim light (CD) | Warm, bright light (WB) | Warm, dim light (WD) |
| --- | --- | --- | --- | --- |
| S-cone-opic (E$^{D65}_{v,sc}$ in lux) | 900 | 90 | 268 | 27 |
| M-cone-opic (E$^{D65}_{v,mc}$ in lux) | 974 | 98 | 777 | 76 |
| L-cone-opic (E$^{D65}_{v,lc}$ in lux) | 1001 | 100 | 1021 | 99 |
| Rhodopic (E$^{D65}_{v,r}$ in lux) | 887 | 89 | 477 | 47 |
| Melanopic (E$^{D65}_{v,mel}$ in lux) | 853 | 86 | 386 | 39 |
| Photopic (E$_v$ in lux) | 1012 | 101 | 1004 | 98 |
| CCT (K) | 5880 | 5890 | 2676 | 2722 |

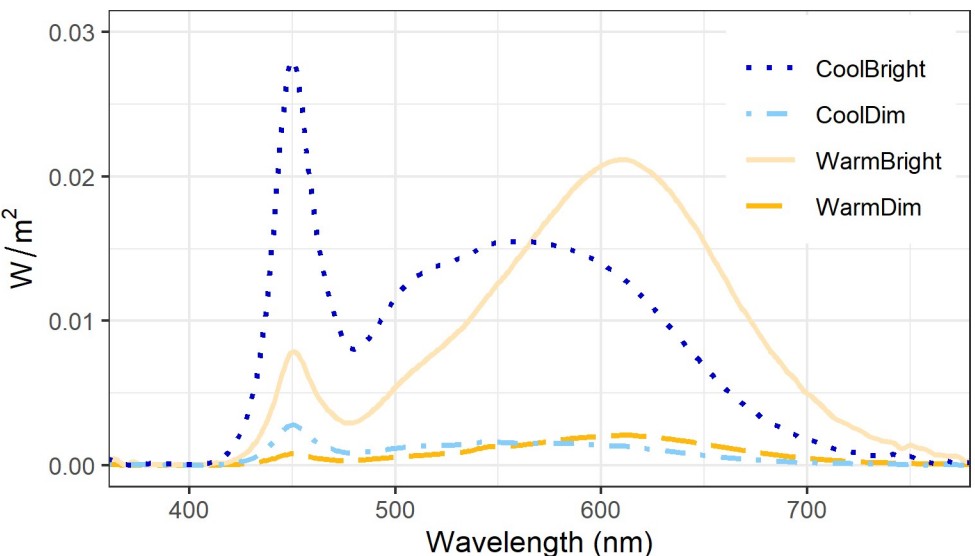

**Fig 2. Spectral power distributions of the light settings.**

Thermal experience was probed with three items, which were based on ASHRAE Standard 55 [52]. Thermal sensation (Sensation_T) was evaluated on a 7-point scale ranging from *Cold (-3)* to *Hot (3)*, thermal acceptance (Acceptance_T) on a binary scale (*Acceptable/Unacceptable*), and thermal comfort (Comfort_T) on a 6-point rating scale ranging from *Very Uncomfortable (-2)* to *Just Uncomfortable (0)* and from *Just Comfortable (1)* to *Very Comfortable (3)*. Additionally, self-assessed shivering (SAS) was evaluated on a VAS ranging from *Not at all (1)* to *Yes, I shiver (10)* as was used in te Kulve, Schlangen, & van Marken Lichtenbelt [50].

Subjective sleepiness was measured using the Karolinska Sleepiness Scale (KSS) with a response scale ranging from *Extremely alert (1)* to *Extremely sleepy (9)* [53]. Six mood-related items (Lively, Awake, Sleepy, Drowsy, Tense and Calm) were selected from the activation–deactivation adjective checklist [54]. Additionally, we included the items Happy and Sad based on prior work [30,55]. Vitality was calculated using the items Lively, Awake, Sleepy, and Drowsy (Cronbach's $\alpha$ = 0.82) that were evaluated on 5-point scales ranging from *Definitely not (1)* to *Definitely (5)*. Vitality was calculated using the factor loadings that were derived through Principal Component Analysis (.70\*Lively + .86\*Awake—.82\*Sleepy—.84\*Drowsy). Participants evaluated their mood state (Tense, Calm, Sad, and Happy) on identical response scales. Tense and Sad were excluded from the analysis as the variance in the responses was too low. Calm was recoded into three categories (recode key: 1–3 = 1, 4 = 2 and 5 = 3) to correct for the skewed response distribution.

**Performance tasks.** In terms of performance we measured executive functioning and vigilance components. The four-minute Backward Digit Span Task (BDST) was used as an indicator of executive functioning [56]. Participants were presented with auditory sequences of numbers that ranged from four to eight digits. They completed two trials per digit-span length, resulting in ten trials per measurement block. After hearing the digits, participants typed the sequence in reversed order, within a time limit (2s + 2.3s per digit). No performance feedback was given. The total number of correct responses was used as the measure for executive functioning.

A five-minute auditory Psychomotor Vigilance Task (PVT) was used as a measure of sustained attention [57]. Participants were asked to keep their dominant rested on the space bar

and respond as fast as possible by pressing the space bar to short auditory stimuli of 400Hz. Stimuli were presented without a prior warning signal and lasted 200ms. The inter-stimulus intervals (ISI) were randomly chosen and ranged between 6 and 25s. Stimuli followed each other immediately with only the ISI in between (i.e., without inter trial interval) and no performance feedback was given. Each stimulus without a key response was coded as missing and counted as a lapse. Responses that deviated more than three standard deviations from the mean were identified per measurement block in each session. These were excluded during post-processing and counted as additional lapses. The mean reaction time in ms of the valid responses during these 5 minutes was used as dependent measure for sustained attention.

**Physiological measurements.** Electrocardiography and electrodermal activity (EDA) measurements were done using TMSi software to collect physiological indicators of arousal. For heart rate (HR) and heart rate variability (HRV), participants attached one electrode on the left clavicula, one on the soft tissue below the right clavicula and one on the soft tissue right below the ribs on the left side of the body. The EDA electrodes were attached to the first phalanxes of the middle and ring finger of the non-dominant hand to measure skin conductance level (SCL) in micro Siemens. Institutional software was used for outlier and artefact detection [58–60]. Mean HR, HRV (determined as root mean square of successive differences) and SCL during the five-minute PVT were used in the analyses.

Sixteen iButton (DS1925) dataloggers (sample interval 300s) attached using Fixomull Stretch tape were used to measure bodily temperatures. The iButtons were placed on 14 ISO-defined body sites [61], based on which mean skin temperature ($T_{average}$) was calculated. Additionally, iButtons were placed at the under arm and the middle finger to assess peripheral vasoconstriction [62]. The proximal skin temperature ($T_{proximal}$) was computed by averaging the temperature measured at the scapula, paravertebral, upper chest, and abdomen. The distal skin temperature ($T_{distal}$) was computed by averaging the fingertip, instep, hand, and forehead skin temperature. To avoid a disproportional distribution, fingertip and hand temperatures were averaged first. Mean skin temperature and the DPG ($T_{distal}$–$T_{proximate}$) during the five-minute PVT were calculated to assess thermoregulation.

**Control measurements.** Light exposure on the day of the session was measured from awakening until the start of the session using a light sensor (LightLog). Furthermore, the Core Consensus Sleep Diary [63] was administered to assess sleep timing, duration and quality of the night before the experimental session. In addition, information on behavior, caffeine and food consumption was gathered at the start of the session (see S1). During the session, participants additionally evaluated the effort they had invested in the reading task and both performance tasks on a visual analogue scales (VAS) ranging from *None (0)* to *Very Much (20)*.

## Procedure

The study was conducted between May 24$^{th}$ and June 24$^{th}$, 2019. Participants completed all experimental sessions at the same time of day (8:45, 10:45, 13:30 or 15:30) to overcome time-of-day effects within participants. The sessions were generally scheduled at least one day apart to avoid session-to-session effects. Two participants had two of the four sessions on subsequent days due to practical limitations in the planning. Participants' clothing was standardized at an estimated clothing value of .7 clo, including insulation of the chair.

After welcoming the participants, the adaptation period of 30 minutes in warm, dim light in the mild cold environment started. While adapting to this environment, control measures were taken and participants were instructed. Subsequently, participants applied the sensors for the physiological measurements and they practiced both the PVT and the BDST. After this adaptation period, the physiological measurements were started and continued for 60 minutes

until the end of the session. During this period, the self-report measures and performance tasks were sampled every 15-min. The self-reports were completed in on average 84 ± 35s, after which the five-minute PVT and the four-minute BDST were performed. The first measurement, always in warm, dim light, was used as a baseline measurement. In the spare time throughout the procedure, participants read a book. The procedure, as shown in Fig 1, was identical for all sessions. At the end of the fourth session, participants were thanked, debriefed and compensated.

## Statistical analysis

MATLAB R2017b was used for all data processing and RStudio 1.1.463 for all analyses. The 'psych' and 'plyr' packages were used for the preparatory analyses; for the statistical analysis the packages 'emmeans', 'Hmisc', 'lme4' and 'lmerTest' were used. The 'ggplot2'-package was used for all visualizations. Outliers were identified by looking at the normal distributions and removing cases that deviated more than three standard deviations from the mean. From the binary variables percentual scores were calculated and visual inspections were done instead of statistical analyses.

The main analyses examined the effect of *Illuminance* and *CCT* over time after the abrupt transition in lighting. In these mixed model analyses, Participant (P) and Session (S; nested within Participant) were added as random intercepts within the model. The models for each of the different dependent variables further included *CCT* (Warm/Cool), *Illuminance* (Bright/ Dim), *Measurement block* (1/2/3), the two-way interactions *CCT\*Illuminance*, *CCT\*Block* and *Illuminance\*Block*, the three-way interaction *CCT\*Illuminance\*Block* and *Time of day* (morning/afternoon) as fixed predictor variables. The *Baseline score* was added to account for potential baseline differences and *Reading effort* as an indicator of participants' effort spent in between assessments. No other control variables were used as no statistically significant differences existed between neither the experimental conditions nor morning vs. afternoon sessions in the control variables assessed at session level. Furthermore, we examined whether random slopes for either Illuminance or CCT at the participant level significantly improved the model to examine the existence of significant interindividual differences in the responses. Only in case the addition of the random slope resulted in a significant improvement compared to the model without random slope as determined by a likelihood-ratio test, the random slope was added to the final model. The model was specified as follows: $Y_{ijk} = \beta_0 + P_{00k} + S_{0j} + \beta_1 \cdot Block_{ijk} + (\beta_2 + P_{2k}) \cdot CCT_{jk} + (\beta_3 + P_{3k}) \cdot Illuminance_{jk} + \beta_4 \cdot CCT_{jk} \cdot Illuminance_{jk} + \beta_5 \cdot CCT_{jk} \cdot Block_{ijk} + \beta_6 \cdot Illuminance_{jk} \cdot Block_{ijk} + \beta_7 \cdot CCT_{jk} \cdot Illuminance_{jk} \cdot Block_{ijk} + \beta_8 \cdot TimeOfDay_k + \beta_9 \cdot Y_{baseline,jk} + \beta_{10} \cdot ReadingEffort_{ijk} + e_{ijk}$. Based on the models, contrast analyses were performed to examine the onset and persistence of the effect of *Illuminance* and *CCT* separately per measurement block. For all analyses, an $\alpha$ of 0.01 was used as cut-off for statistical significance.

## Results

Onset and persistence of the effects of illuminance and CCT were tested for all dependent variables using the repeated measurements within the sessions. Existence of structural interindividual variability in the responses to transitions in illuminance and CCT existed was investigated with random slope models. For conciseness, we report only the estimated marginal means and coefficients of the significant parameters in text. Additionally, the statistics for the main predictors (*Illuminance*, *CCT*, *Block*, *Illuminance* x *CCT*, *Illuminance* x *Block* and *CCT* x *Block*) for all dependent variables are presented in Table 3. Further statistics for the other predictors (*TimeOfDay*, *ReadingEffort*, $Y_{baseline}$ and *CCT* x *Illuminance* x *Block*), the full model and the null model for all dependent variables are presented in the Supplementary

**Table 3. Statistics for the main predictors for all outcome parameters.**

| Dependent variable | Illuminance | | CCT | | Block | | Illuminance x CCT | | Illuminance x Block | | CCT x Block | |
|---|---|---|---|---|---|---|---|---|---|---|---|---|
| | $F$ | $p$ | $F$ | $p$ | $F$ | $p$ | $F$ | $p$ | $F$ | $p$ | $F$ | $p$ |
| Sensation$_{VI}$ | **$F_{1,65}=145.79$** | **< 0.001** | $F_{1,66}=4.25$ | 0.04 | **$F_{2,182}=13.46$** | **< 0.001** | $F_{1,66}=5.75$ | 0.02 | **$F_{2,181}=11.15$** | **<0.001** | $F_{2,182}=0.10$ | 0.91 |
| Sensation$_{vc}$ | $F_{1,69}=0.00$ | 1.00 | **$F_{1,69}=90.16$** | **< 0.001** | $F_{2,181}=0.88$ | 0.42 | $F_{1,69}=0.00$ | 0.94 | $F_{2,181}=0.26$ | 0.77 | $F_{2,181}=16.83$ | **< 0.001** |
| Comfort$_V$ | $F_{1,68}=0.31$ | 0.58 | **$F_{1,68}=7.79$** | **< 0.01** | **$F_{2,183}=4.85$** | **< 0.01** | $F_{1,68}=3.36$ | 0.07 | $F_{2,183}=1.32$ | 0.27 | $F_{2,183}=4.83$ | < 0.01 |
| Vitality | **$F_{1,66}=10.78$** | **< 0.01** | $F_{1,66}=0.48$ | 0.49 | $F_{2,182}=3.87$ | 0.02 | $F_{1,67}=0.95$ | 0.95 | $F_{2,182}=1.89$ | 0.15 | $F_{2,182}=1.13$ | 0.32 |
| Sleepiness (KSS) | **$F_{1,67}=8.24$** | **< 0.01** | $F_{1,67}=0.18$ | 0.67 | $F_{2,183}=3.75$ | 0.03 | $F_{1,69}=0.22$ | 0.64 | $F_{2,183}=1.97$ | 0.14 | $F_{2,183}=1.38$ | 0.25 |
| Mean RT (PVT) | $F_{1,48}=0.37$ | 0.54 | $F_{1,47}=0.72$ | 0.40 | $F_{2,153}=0.08$ | 0.92 | $F_{1,48}=0.21$ | 0.65 | $F_{2,151}=0.06$ | 0.94 | $F_{2,152}=3.52$ | 0.03 |
| Effort PVT | $F_{1,59}=1.59$ | 0.21 | $F_{1,58}=0.04$ | 0.84 | $F_{2,172}=1.16$ | 0.31 | $F_{1,60}=0.12$ | 0.73 | $F_{2,171}=1.72$ | 0.18 | $F_{2,172}=1.18$ | 0.31 |
| Correct (BDST) | $F_{1,65}=0.00$ | 1.00 | $F_{1,65}=0.15$ | 0.70 | $F_{2,183}=1.91$ | 0.15 | $F_{1,67}=0.03$ | 0.85 | $F_{2,183}=0.47$ | 0.62 | $F_{2,183}=1.31$ | 0.27 |
| Effort BDST | $F_{1,67}=6.58$ | 0.01 | $F_{1,66}=0.06$ | 0.80 | $F_{2,182}=0.52$ | 0.60 | $F_{1,68}=1.82$ | 0.18 | $F_{2,182}=1.65$ | 0.20 | $F_{2,182}=0.02$ | 0.98 |
| Mean SCL | $F_{1,62}=0.02$ | 0.88 | $F_{1,62}=0.66$ | 0.42 | **$F_{2,164}=18.48$** | **< 0.001** | $F_{1,63}=1.52$ | 0.22 | $F_{2,163}=2.17$ | 0.12 | $F_{2,164}=0.96$ | 0.38 |
| Mean HR | $F_{1,65}=1.68$ | 0.20 | $F_{1,65}=1.75$ | 0.19 | $F_{2,167}=1.75$ | 0.18 | $F_{1,67}=0.32$ | 0.58 | $F_{2,167}=1.03$ | 0.36 | $F_{2,167}=1.27$ | 0.28 |
| Mean HRV | $F_{1,86}=0.88$ | 0.35 | $F_{1,86}=1.82$ | 0.18 | **$F_{2,169}=28.76$** | **< 0.001** | $F_{1,87}=0.09$ | 0.76 | $F_{2,169}=0.13$ | 0.88 | $F_{2,169}=1.14$ | 0.32 |
| Calm | $F_{1,69}=0.16$ | 0.69 | $F_{1,67}=0.31$ | 0.26 | $F_{2,184}=0.96$ | 0.39 | $F_{1,69}=0.15$ | 0.70 | $F_{2,183}=1.03$ | 0.36 | $F_{2,183}=0.67$ | 0.51 |
| Happy | $F_{1,66}=0.32$ | 0.58 | $F_{1,65}=3.20$ | 0.08 | $F_{2,180}=2.62$ | 0.08 | $F_{1,65}=0.06$ | 0.81 | $F_{2,180}=2.23$ | 1.11 | $F_{2,180}=3.38$ | 0.04 |
| Sensation$_T$ | $F_{1,67}=0.37$ | 0.55 | $F_{1,67}=0.20$ | 0.65 | **$F_{2,182}=11.76$** | **< 0.001** | $F_{1,68}=0.11$ | 0.75 | $F_{2,181}=0.52$ | 0.60 | $F_{2,181}=0.86$ | 0.43 |
| Self-assessed shivering | $F_{1,61}=0.02$ | 0.90 | $F_{1,61}=1.42$ | 0.24 | **$F_{2,177}=9.16$** | **< 0.001** | $F_{1,62}=0.57$ | 0.45 | $F_{2,177}=0.41$ | 0.66 | $F_{2,177}=0.49$ | 0.61 |
| Comfort$_T$ | $F_{1,63}=1.95$ | 0.17 | $F_{1,63}=3.41$ | 0.07 | $F_{2,179}=4.07$ | 0.02 | $F_{1,64}=0.07$ | 0.80 | $F_{2,178}=1.02$ | 0.36 | $F_{2,179}=0.50$ | 0.61 |
| T$_{skin}$ | $F_{1,69}=0.43$ | 0.51 | $F_{1,69}=1.29$ | 0.26 | **$F_{2,181}=298.73$** | **< 0.001** | $F_{1,69}=6.50$ | 0.01 | $F_{2,181}=0.18$ | 0.84 | $F_{2,181}=0.03$ | 0.97 |
| DPG | $F_{1,69}=1.19$ | 0.28 | $F_{1,69}=0.04$ | 0.83 | **$F_{2,183}=22.54$** | **< 0.001** | $F_{1,70}=0.21$ | 0.65 | $F_{2,182}=2.20$ | 0.11 | $F_{2,183}=0.42$ | 0.66 |

The statistics presented bold represent statistically significant effects. *F*: F-statistic and *p*: p value.

materials S1 and S2 Tables. Additionally, S3 Table provides the contrast estimates for the main effects of Illuminance and CCT and S4 Table reports on the contrast estimates for Illuminance and CCT per measurement block. Significance of the latter is also indicated in Figs 3 and 5–9 using an asterisk (*) for significant effects (p < 0.01) for illuminance and a plus (+) for CCT.

## Visual experience

Participants' sensation of the brightness of the light (Fig 3A) was significantly affected by *Illuminance*. As expected, the bright conditions were perceived significantly brighter (*Estimated Marginal Mean* [*EMM*] ± *Standard Error* [*SE*] = 1.36 ± 0.13) than the dim conditions

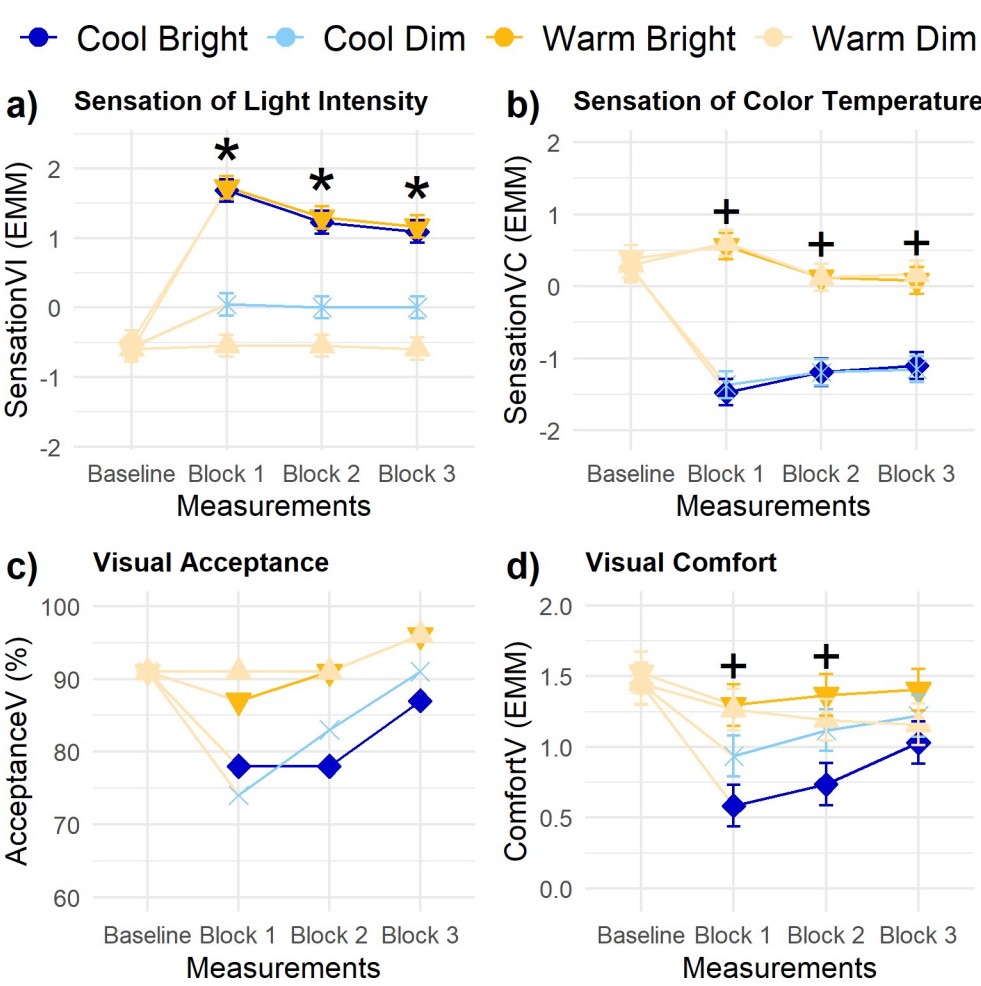

**Fig 3. Trajectories of visual experience parameters.** (A) sensation of light intensity, (B) sensation of color temperature, (C) visual acceptance (in %—no statistical testing), and (D) visual comfort. Error bars are standard errors (SE). Contrasts for illuminance and CCT were done for each measurement block: + indicates p < .01 for CCT, * p < .01 for illuminance.

(-0.28 ± 0.13). Additionally, there was a significant main effect of *Measurement block* and an interaction of *Illuminance* and *Block* (see Table 3). Over the three measurement blocks, the brightness sensation of the bright conditions converged slightly towards the brightness sensation of the dim conditions. Yet, Fig 3A shows that the effect of *Illuminance* emerged right after the transition and persisted throughout the remaining two measurement blocks. Participants' sensation of the color of the white light (Fig 3B) was significantly affected by *CCT*. In line with the expectations, the warm conditions were perceived as warmer (0.29 ± 0.14) compared to the cool light conditions (-1.24 ± 0.14). In addition, there was a significant interaction effect of *CCT* and *Block*, reflecting a slightly diminishing difference between warm and cool conditions in the second and third measurement block compared to the first. Still, a statistically significant difference in sensation of the color of the light persisted across all measurement blocks as can be seen in Fig 3B.

Visual inspection of the results for visual acceptance suggest a lower visual acceptance of the light conditions after transitioning to the cool light conditions compared to both warm light conditions (Fig 3C). Over time, the percentage of persons rating the light setting as

acceptable increased. For visual comfort, there was a statistically significant main effect of *CCT* (Fig 3D), but not of *Illuminance*. However, the effect of *Illuminance* on Comfort$_V$ was structurally influenced by interindividual differences ($\chi^2$ (2) = 18.82, p<0.001); the change in illuminance of the light led to widely varying visual comfort votes by the participants (range = -0.50 to 1.57). Fig 4A clearly shows that the bright light was evaluated as more comfortable than the dim light by some participants, but less comfortable by others. In contrast, the effect of *CCT* on Comfort$_V$ showed no statistically significant interindividual differences in responsiveness ($\chi^2$ (2) = 0.26, p = 0.88). Fig 4B indicates that the difference in CCT of the light conditions led to comparable changes in visual comfort evaluations for all participants; the warm light conditions were, on average, experienced as more comfortable (1.27 ± 0.10) compared to the cool light conditions (0.92 ± 0.10). Fig 3D shows that the effect of *CCT* on Comfort$_V$ was only statistically significant in the first two measurement blocks after the transition. Last, *Reading effort* ($\beta$ = -0.03 ± 0.01) and *Baseline visual comfort* ($\beta$ = 0.28 ± 0.09) had a significant effect on the visual comfort of the light condition. Comfort$_V$ decreased with a higher self-reported reading effort.

## Vitality, sleepiness and performance

The significant main effect of *Illuminance* on vitality (Fig 5A) indicated that vitality was, on average, higher in the bright light conditions (1.18 ± 0.29) compared to the dim light conditions (0.18 ± 0.29). However, the contrast analyses revealed that bright light led to significantly increased vitality in the first two measurement blocks after the transition only. Random slope analyses showed no statistically significant interindividual differences. Vitality was significantly influenced by *Reading effort* ($\beta$ = -0.19 ± 0.03) and *Baseline level of vitality* ($\beta$ = 0.40 ± 0.07). Vitality decreased with increasing self-reported reading effort. The response pattern for sleepiness was highly similar, but–as expected–reversed (Fig 5B). *Illuminance* had a statistically significant main effect on sleepiness, indicating that the bright conditions were related to a lower sleepiness (4.57 ± 0.21) compared to the dim light conditions (5.22 ± 0.21). Again, this effect was only significant in the first two measurement blocks. Sleepiness was additionally affected by *Reading effort* ($\beta$ = 0.11 ± 0.02) and the level of sleepiness at baseline ($\beta$ = 0.33 ± 0.08). Sleepiness increased with a higher self-reported reading effort.

**Fig 4. Individual differences in the visual comfort evaluation.** Individual values of visual comfort are calculated using the $EMM + P_{00k} + P_{3k} * x$ based on the model with random slopes for (A) Illuminance and (B) CCT.

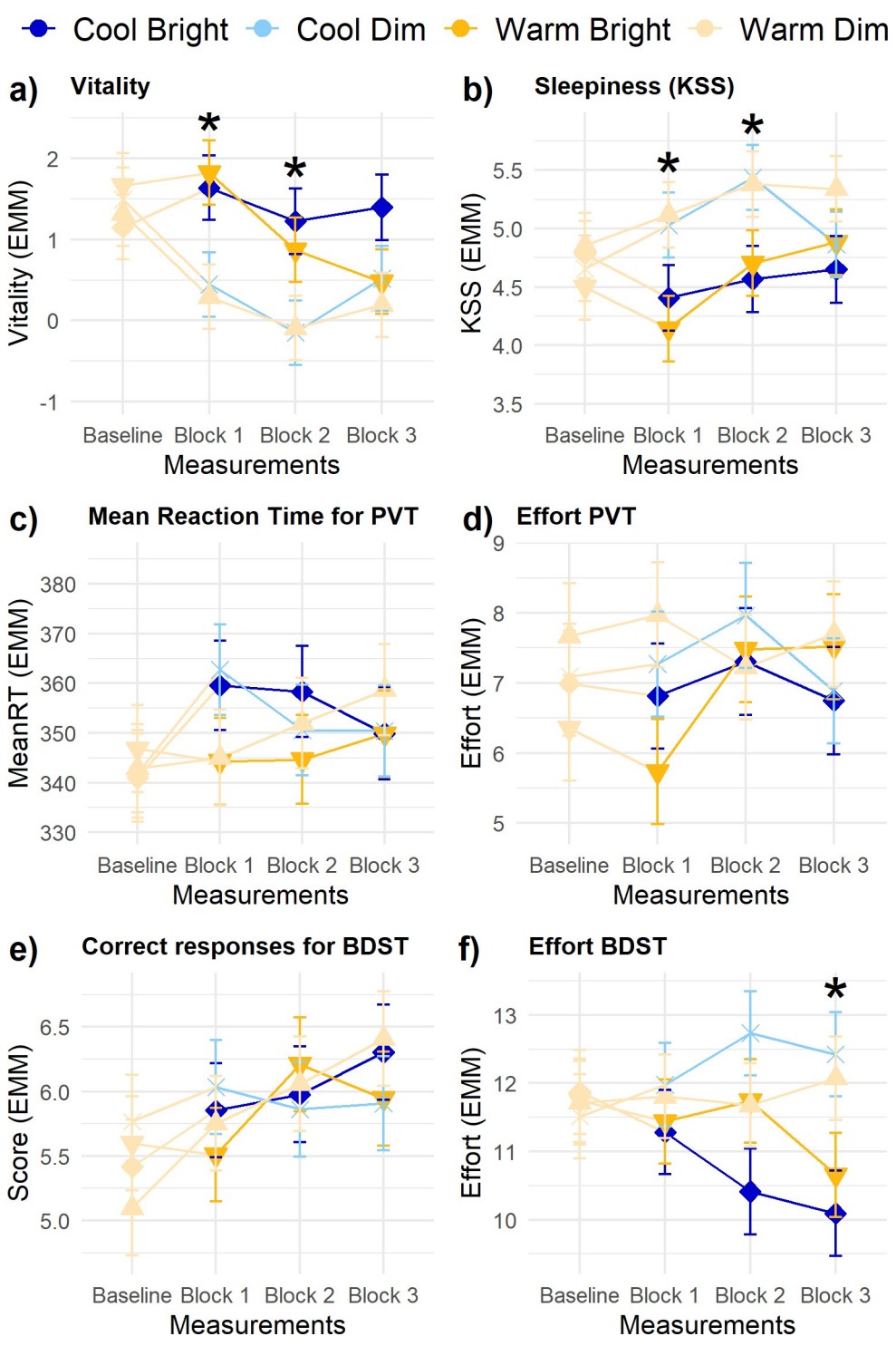

**Fig 5. Trajectories of vitality, sleepiness and performance parameters.** (A) vitality, (B) sleepiness, (C) mean reaction time for PVT, (D) effort during the PVT, (E) correct responses for BDST, and (F) effort during the BDST. Error bars are *SE*. Contrasts for illuminance and CCT were done for each measurement block: + indicates $p < .01$ for CCT, * $p <$ .01 for illuminance.

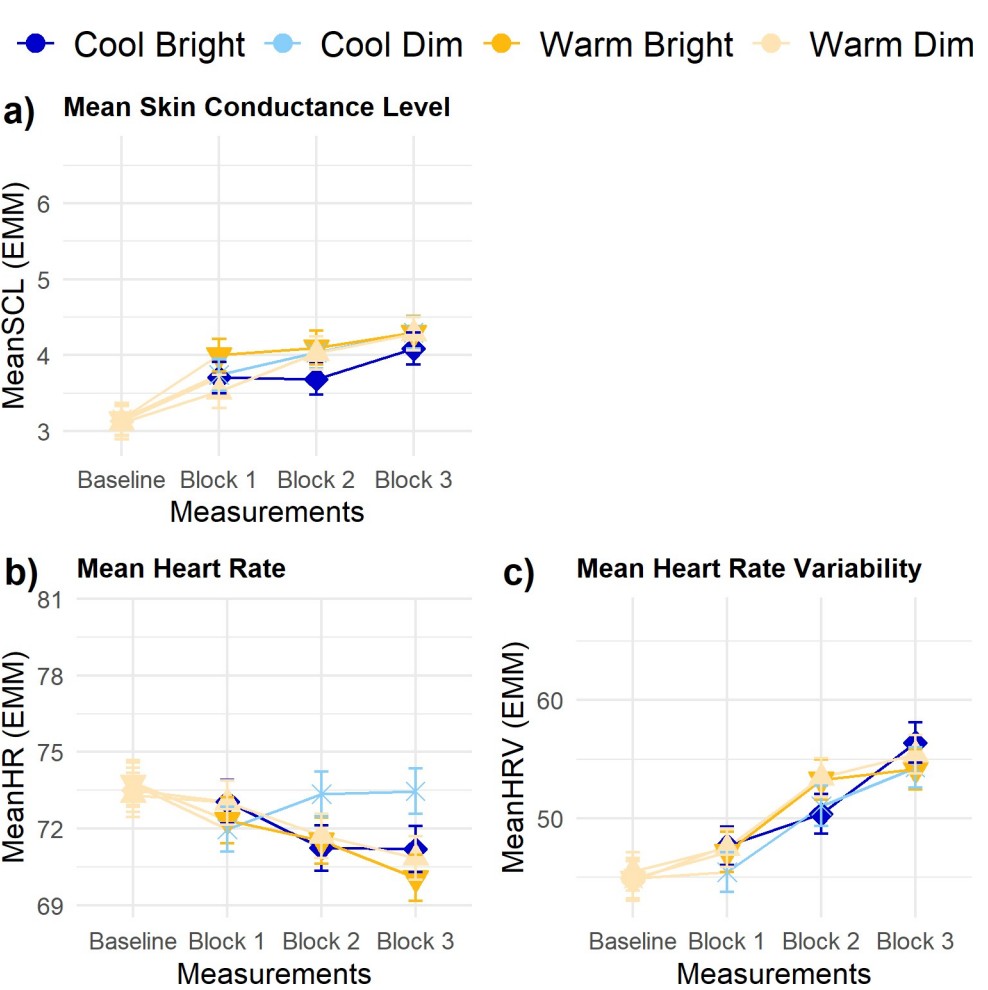

**Fig 6. Trajectories of physiological parameters.** (A) mean SCL, (B) mean HR, and (C) mean HRV. Error bars are SE. No significant differences existed for illuminance and CCT in any of the measurement blocks.

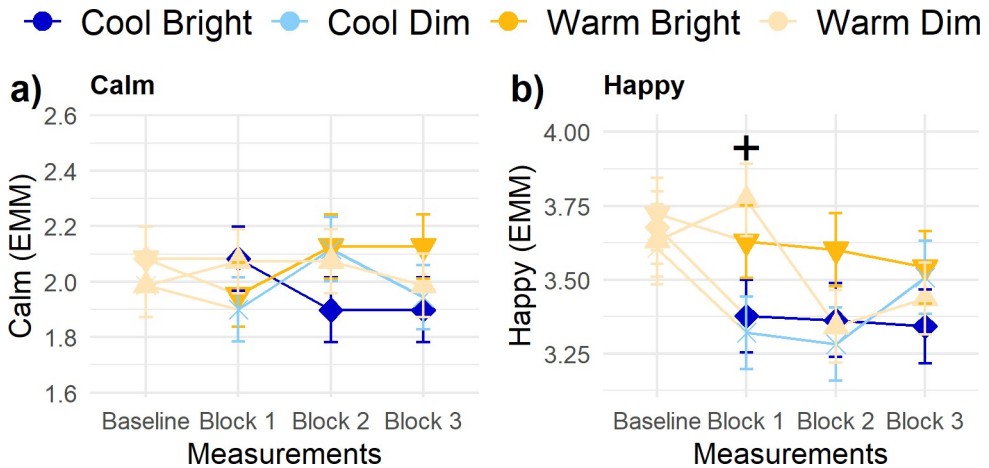

**Fig 7. Trajectories of mood parameters.** (A) Calm, and (B) Happy. Error bars are SE. Contrasts for illuminance and CCT were done for each measurement block: + indicates p < .01 for CCT, * p < .01 for illuminance.

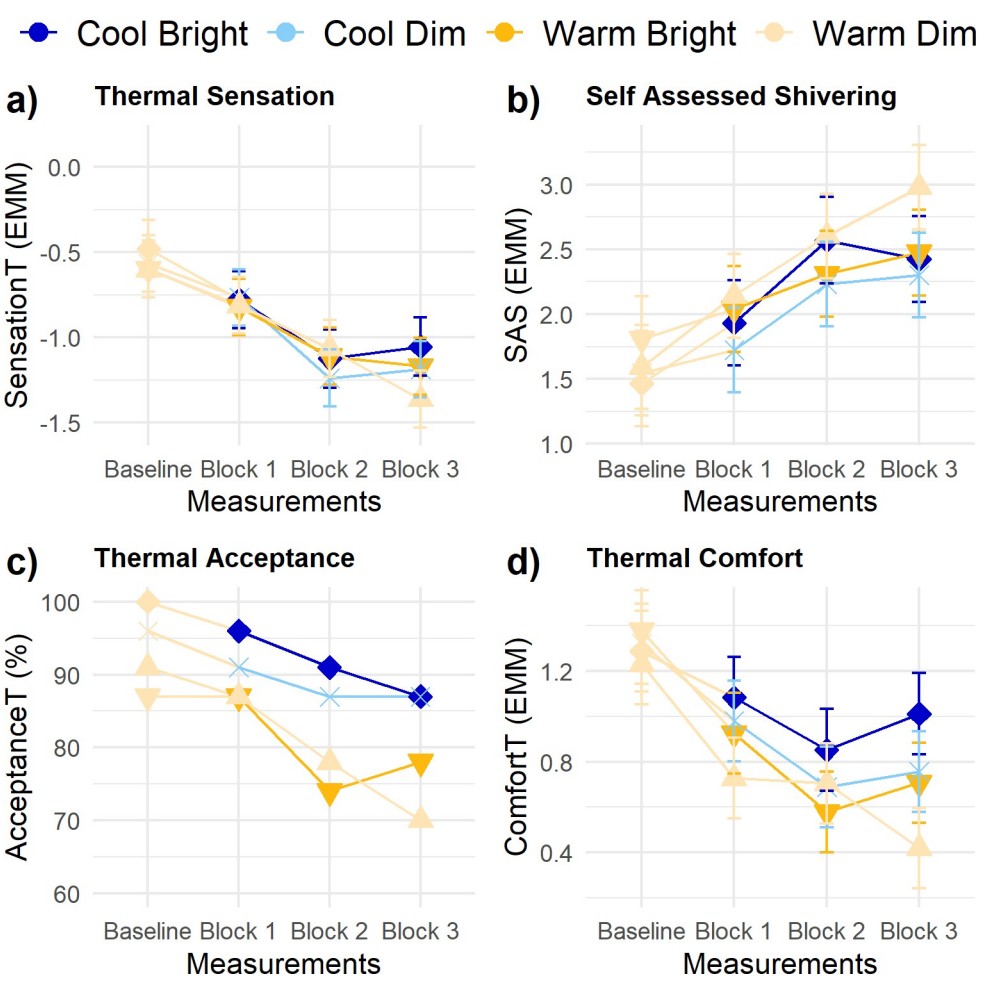

**Fig 8. Trajectories of thermal experience parameters.** (A) thermal sensation, (B) self-assessed shivering, (C) thermal acceptance (in %—no statistical testing), and (D) thermal comfort. Error bars are SE. No significant differences existed for illuminance and CCT in any of the measurement blocks.

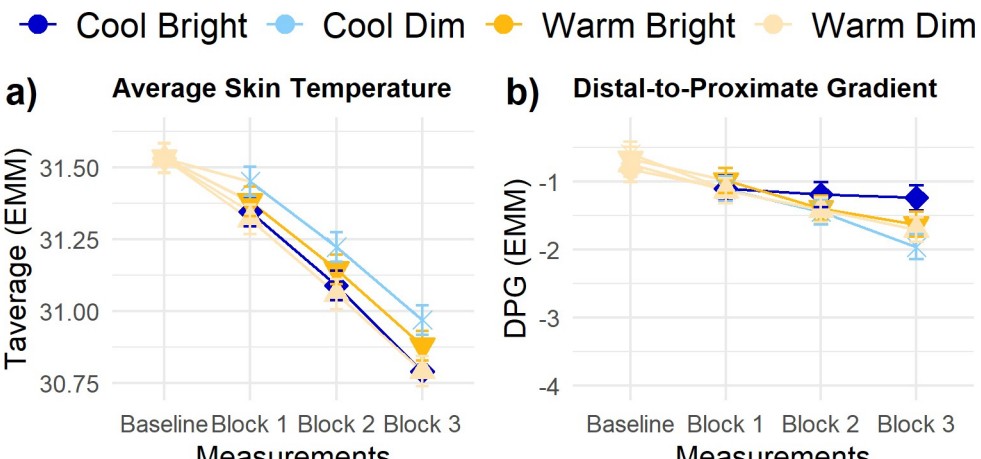

**Fig 9. Trajectories of thermoregulation related parameters.** (A) average skin temperature, and (B) distal-to-proximate gradient. Error bars are *SE*. No significant differences existed for illuminance and CCT in any of the measurement blocks.

The response time during the PVT showed neither statistically significant main nor interaction effects of *Illuminance* and *CCT* (Fig 5C), nor interindividual differences of these effects. *Baseline response time* significantly predicted the response time on the PVT (β = 0.51 ± 0.06). Similarly, *Baseline BDST score* was a significant predictor for the number of correct responses on the BDST (Fig 5E; β = 0.35 ± 0.05). No main or interaction effect of *Illuminance* and *CCT* on the score of the BDST occurred in any of the measurement blocks. Also for the self-reported exerted effort during the tasks no main or interaction effects of *Illuminance* and *CCT* were found (Fig 5D and 5F). The effort exerted in the PVT was only significantly affected by *Reading effort* (β = 0.33 ± 0.05) and the effort during the baseline measurement (β = 0.35 ± 0.07). For the effort exerted in the BDST only the effort during the baseline measurement was statistically significant (β = 0.71 ± 0.05). Furthermore, in the third measurement block, the effort exerted in the BDST was significantly lower in the bright light compared to the dim light (Fig 5F).

## Physiological arousal

SCL, HR and HRV were not significantly influenced by the light conditions (Fig 6A–6C) nor by interindividual differences in the response to these light conditions. All three parameters were significantly affected by their *Baseline level* (β = 1.02 ± 0.05; β = 0.89 ± 0.03 and β = 0.87 ± 0.05, respectively). SCL and HRV were, additionally, significantly influenced by *Block*, both showing an increase over time in the experimental session.

## Mood

Participants' calmness ratings were significantly influenced by *Baseline level* (β = 0.50 ± 0.07) and the three-way interaction of *Illuminance*, *CCT* and *Block* suggesting slightly different trajectories for all conditions (Fig 7A). Yet, no main and interaction effects of *Illuminance* and *CCT* in these responses occurred in any of the measurement blocks. Happiness showed an significant effect of its baseline score (Fig 7B; β = 0.61 ± 0.08). In the first measurement block, there was also a significant difference between the warm and cool conditions: the transition to cool light led to a decrease in reported happiness in the first measurement block. These effects turned insignificant in the following two measurement blocks.

## Thermal experience

Thermal sensation of the participants (Fig 8A) was significantly influenced by *Block* and *Baseline thermal sensation* (β = 0.35 ± 0.09). Sensation$_T$ decreased over time, but in none of the measurement blocks a significant main or interaction effect of *Illuminance* or *CCT* occurred. Similarly, self-assessed shivering (Fig 8B) was only significantly affected by *Block* and *Baseline level* (β = 0.46 ± 0.13). Self-assessed shivering increased over time in session. Visual inspection of the percentage of persons rating the lighting conditions as thermally acceptable (Fig 8C) suggested a decrease over time in all conditions. The visual inspection also suggested a lower thermal acceptance in the warm light conditions, however, this difference already seemed to exist in the baseline. Comfort$_T$ (Fig 8D) was significantly affected by *Baseline level* (β = 0.21 ± 0.08) and *Reading effort* (β = -0.04 ± 0.01). Participants' thermal comfort significantly decreased over time and with increasing reading effort. The thermal experiences were not affected by the changes in the light conditions and showed no statistically significant interindividual variability in their responses to the lighting conditions.

## Thermoregulation

For the average skin temperature, there was no statistically significant main or interaction effect of *Illuminance* and *CCT* in any of the measurement blocks (Fig 9A), nor were there statistically significant interindividual differences in these effects. The average skin temperature was significantly affected by *Block* and *Baseline skin temperature* ($\beta = 0.98 \pm 0.03$). Average skin temperature decreased over time. Similarly, the DPG (Fig 9B) was significantly influenced by *Block* and *Baseline DPG* ($\beta = 1.14 \pm 0.03$). Over the three measurement blocks, the DPG decreased (i.e., the difference between distal and proximal temperatures increased).

## Discussion

With this laboratory study, we aimed to study the impact of an abrupt transition in illuminance, in CCT, and the interaction thereof, on temporal trajectories of alertness, arousal, comfort and mood. Furthermore, we investigated the presence of interindividual variability in these temporal response dynamics. Healthy participants were exposed to four experimental light conditions and repeatedly completed subjective measures of comfort, alertness and mood, and performance measures probing vigilance and executive functioning, while continuously tracking physiological indicators of arousal and thermoregulation. We largely replicated findings reported by Kompier et al. [28], but–due to the adapted design of the current study–effects could now be uniquely attributed to changes in illuminance or CCT. Additionally, no statistically significant interaction effects of CCT and illuminance were found for any of the parameters. The extent to which the abrupt transitions in illuminance and/or CCT influenced subjective evaluations of comfort and mood as well as subjective and objective measures of alertness, arousal and thermoregulation over time can be described in terms of the onset and persistence of the response trajectory.

### Onset-acute effects of an abrupt transition

The abrupt transitions in illuminance and CCT led to an immediate change in brightness and color sensation. Whereas the change in brightness perception could be clearly attributed to change in illuminance of the light, both color sensation and visual comfort showed a main effect of the transition in CCT only. More specifically, participants reported diminished visual comfort directly after the transition to cool light compared to the warm light conditions. Furthermore, participants reported feeling less happy right after transitioning to cool light compared to warm light. Significant decreases in sleepiness and increases in vitality also occurred immediately after the transition to bright light compared to the continuous dim light, which is in line with prior research [28,30]. In contrast, none of the task performance and physiological arousal parameters showed either a direct or delayed response within 45 minutes after the abrupt transition in illuminance and/or CCT. A potential reason for the lack of response to the light conditions in terms of task performance could be the compensatory influence of the exerted effort on the task [64]; in some conditions the light might have resulted in better performance, but in the conditions in which light did not positively affect the performance, participants may have exerted more effort and thereby compensated for the absent effect of light. For both the PVT and the BDST the self-reported exerted effort was indeed slightly lower in bright light compared to dim light, albeit not statistically significantly so. This emphasizes the importance of including assessments of the effort or task engagement for performance tasks. Additionally, the analysis of other parameters such as lapses, standard deviation or the variability of the RT [65] during the PVT may yield more insight in the responses of the participants during the PVT, and the potential shift in response as a consequence of the light transitions or the time in session. To ensure sufficient variation in these parameters, a PVT with more trials

(e.g., achieved by having a longer duration of the task in total or shorter ISIs within the task) than the PVT version employed in the current study would be required. With respect to longer performance tasks, it is important to realize that simple reaction time tasks can affect participants by inducing mental fatigue and strain [66]. Longer tests should therefore preferably also be accompanied by additional subjective state measurements such as task engagement, distress, and worry to examine how performing such a task affects subjective state and to be able control for this. Yet, the absence of effects of different light conditions on performance and physiological arousal measures is not uncommon, and prior studies have reported that effects on these measures may depend on certain times of day, exposure duration, task difficulty, or season [22,25,29,55]. Although this could also explain why light did not affect these responses in the present study, we should question whether the acute effects of light on performance tasks and physiological arousal are practically relevant if they only occur in specific contexts. Last, in this study no direct evidence for cross-sensory effects of illuminance or CCT on thermal experience was found, although various theories (e.g., hue-heat hypothesis [67], brightness associations [68] or physiological effects [69]) hypothesize such effects. In sum, this study demonstrated that subjective responses to an abrupt transition in illuminance and/or CCT, if they emerge, always emerge virtually immediately and that task performance and physiological measures do not respond within the first 45 minutes after an abrupt transition.

### Persistence–sustained effects of an abrupt transition

Not all effects that occurred right after the transition persisted throughout the entire duration of exposure. Only the sensation of the color and brightness of the light remained significantly different between conditions across all three measurement blocks, even though the difference diminished slightly. As in Kompier et al. [28], the contrast between the persistent response for the visual sensation and the transient response for visual comfort and acceptance can be explained by adaptation to the light setting. Although participants' sensation of the light persisted, the adaptation of the eye likely led to altered acceptance and comfort votes. The abrupt transition to cool light led to an immediate decrease in visual comfort, but this effect disappeared over time. This is partially in contrast to the review by Fotios et al. [70] who discussed that pleasantness ratings may reach a plateau at 6 minutes after light onset in certain settings, but also demonstrated that—overall—the effects of CCT on ratings of brightness and pleasantness are small and seem largely independent of adaptation time. In contrast, in the current study the visual comfort of the participants improved gradually after the light transition to cool, bright light. This suggests that initially participants may not appreciate an abrupt transition to cool light, but when time passes their visual comfort seems to returns towards the original level. Potentially, more gradual light conditions can prevent this initial transient decline in comfort. Happiness also responded temporarily to the transition in CCT, which was not in line with Kompier et al. [28], who found no effects of a transition from warm, dim to cool, bright lighting on happiness. The effects of the light conditions on vitality and alertness, on the other hand, were in line with earlier studies [24–28]. The effects of illuminance on vitality and alertness persisted, at least throughout the first 20 minutes after the transition. In this study, we demonstrated that the change in illuminance of the light, rather than the change in CCT, was significantly related to a change in sleepiness and vitality. The $E^{D65}_{v,mel}$ of the bright compared to the dim conditions implied a tenfold increase, whereas the $E^{D65}_{v,mel}$ 'merely' doubled from the warm to the cool condition. In line with this, the–nonsignificant–difference in sleepiness and vitality between warm vs. cool light was about one fifth of the size of the effect of illuminance. Rather than attributing the alerting effect to illuminance or CCT per se, the melanopic activation that a light setting can achieve is the more likely explanation for this

acute, alerting effect [71,72], as was also suggested in the study by Ru et al. [41]. In this respect, it is important to note that the factor two difference in $E^{D65}_{v,mel}$ between the warm, bright and the cool, bright light condition did not lead to a difference in sleepiness or vitality, which is possibly due to the nonlinear (sigmoidal) relationship between light and alerting responses [9,72].

## Interindividual variability

In this study, we additionally examined whether significant interindividual variability existed for the main effects of illuminance and CCT. To test the existence of this variability, random slopes for illuminance and CCT were added separately at the participant level in the models. The interindividual variability reported here therefore reflects variation between persons that is stable throughout the period during which participants completed the four sessions. More specifically, this variability is likely trait-like and potentially a function of person characteristics, such as lens characteristics or chronotype. Additionally, variation in responsiveness to light may also emerge due to behavioral and situational differences, such as prior light exposure [47], but these were not examined in the current study. Significant interindividual variability was exclusively found for the effect of illuminance on visual comfort, which might explain the absence of a main effect of illuminance on visual comfort. Both the direction and the extent to which individuals' visual comfort was affected were highly dependent on the individual; the transition to a higher illuminance was considered more comfortable by some and less comfortable by others. Examining the origin of this interindividual variation in responses to light requires between-subject comparisons which were not performed in this study due to statistical power considerations. Although individual variability in preferred lighting conditions has been demonstrated before [43,73], the current study is–to our knowledge–the first to demonstrate that individual variability for the effect of illuminance on visual comfort exist, but not for the effect of CCT. This finding, of course, applies only to the employed target audience, stimulus range, duration and timing, and requires further investigations in order to generalize. No significant interindividual variability in the alerting responses were found, despite suggestions thereof in prior studies [26,42]. This might be explained by the homogeneity in the sample; limited variation in factors such as age and chronotype likely results in little variation in responsiveness to light [74].

## Theoretical reflections and practical implications

In the current study, we successfully disentangled the effects of illuminance and CCT on measures of subjective experiences and comfort. The results indicated that manipulations of illuminance and CCT did not interact in their alerting effects, and effects could be attributed to illuminance only. However, the difference in melanopic activation due to the contrasting light conditions may very well be the driving force behind the significant effect that was found [72]. Although effects of illuminance and CCT on visual sensation were straightforward and consistent between persons, visual comfort showed marked interindividual variability. This means that increases in illuminance and CCT may result in a decline in visual comfort for some people, but have contrasting effects for others. None of the other outcome parameters that were included in this study showed statistically significant interindividual variability in their responses to changes in illuminance or CCT, indicating similar responses to light for the relatively homogenous sample that was included in this study. Furthermore, the absence of effects on performance tasks, physiological arousal and thermal experience indicated that no overall daytime effects of light on these parameters emerged, at least not within 45 minutes of exposure, which is largely consistent with recent reviews [49,75].

The different parameters that relate to alertness, arousal and vigilance showed varying response patterns, demonstrating the nuanced differences between how subjective alertness, physiological arousal and behavioral vigilance can be influenced. This emphasizes that interpretation and explanation of effects of light on these parameters and their underlying mechanisms requires a multi-measure approach [76]. Last, the occurrence of both transient and persistent responses underlines the delicacy in timing of measurements within a measurement protocol and thereby the importance of repeated measurements.

### Limitations and future research

Although this study was conducted in a relatively controlled environment, some limitations could not be overcome. In the climate chamber, two office-like workplaces were created to assure that two participants could participate simultaneously. Despite the use of a partitioning wall, participants may have been aware of each other, which potentially impacted concentration on tasks. Furthermore, the sample was predominantly young and healthy, which limits generalizability. The homogeneous sample may have also resulted in the inability to identify interindividual variability for some outcome parameters as these may depend on factors such as chronotype and age which showed little variation in our participant sample. To be able to generalize the results to the general population, larger and more heterogenous samples should also be included in future studies. Participants completed all sessions at the same time of day to control for variation due to time of day within participants, but interaction effects due to time of day between participants could not be tested due to power considerations. Future studies should investigate the effect of seasonality and time of day on the results [25,55,77] and examine the current effects in more naturalistic settings using longer exposure durations before the results can be applied in a dynamic light scenario for office environments. Last, we suggest that future research investigates more closely how visual comfort evolves over the day, extending the work on discomfort glare [78] to visual comfort, to see whether dynamically changing light conditions throughout the day could be stimulating and lead to increased visual comfort.

### Conclusions

In the current study, we demonstrated that abrupt transitions in illuminance and/or CCT led to different temporal response dynamics for subjectively evaluated appraisal, mood and alertness. Illuminance and CCT did not structurally interact on these markers, but each affected a selection (e.g., vitality and sleepiness were moderated by illuminance and visual comfort by CCT). Responses to the abrupt transitions in illuminance and CCT occurred always immediately and exclusively amongst the subjective markers. No obvious responses were found in objective measures of vigilance or arousal. Although subjective alertness benefitted immediately and for at least 20 minutes from higher light levels, we saw no measurable effect for physiological arousal and task performance. Thermal comfort and thermoregulation were also not significantly influenced by the lighting manipulations. In the visual comfort evaluations (and only there) we detected statistically significant interindividual variability in responses to changes in the intensity of the light. The results indicate that the design of dynamic light scenarios that are beneficial for human alertness and vitality requires tailoring to the individual to create visually comfortable environments.

### Supporting information

**S1 Table. Null model statistics.** Variances of the random intercept models.
(PDF)

**S2 Table. Full model statistics.** F-statistic and p-vales of the covariates, full model fit and $R^2$ of the random intercept models.
(PDF)

**S3 Table. Contrast estimates for main effects.** Delta, t-ratio and p-value for the main effect of illuminance and CCT separately.
(PDF)

**S4 Table. Contrast estimates per measurement block.** Delta, t-ratio and p-value per measurement block for the effect of illuminance and CCT separately.
(PDF)

**S1 File. Start questionnaire.** The questions posed at the start of the session as control measures.
(PDF)

# Acknowledgments

We would like to thank Kay Tuip for his help in gathering the data, the HTI lab support team for the technical assistance of the materials and Wout van Bommel for the technical assistance in the laboratory.

# Author Contributions

**Conceptualization:** Maaike E. Kompier, Karin C. H. J. Smolders, Yvonne A. W. de Kort.

**Data curation:** Maaike E. Kompier.

**Formal analysis:** Maaike E. Kompier.

**Funding acquisition:** Yvonne A. W. de Kort.

**Investigation:** Maaike E. Kompier.

**Methodology:** Maaike E. Kompier, Karin C. H. J. Smolders, Yvonne A. W. de Kort.

**Project administration:** Yvonne A. W. de Kort.

**Software:** Maaike E. Kompier.

**Supervision:** Karin C. H. J. Smolders, Yvonne A. W. de Kort.

**Visualization:** Maaike E. Kompier.

**Writing – original draft:** Maaike E. Kompier, Karin C. H. J. Smolders, Yvonne A. W. de Kort.

**Writing – review & editing:** Maaike E. Kompier, Karin C. H. J. Smolders, Yvonne A. W. de Kort.

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
