## [Decision Letter · Decision Letter 0]

14 Jan 2021

PONE-D-20-36022

Abrupt light transitions in illuminance and CCT result in different temporal dynamics and interindividual variability for sensation, comfort and alertness

PLOS ONE

Dear Dr. Kompier,

Thank you for submitting your manuscript to PLOS ONE. After careful consideration, we feel that it has merit but does not fully meet PLOS ONE’s publication criteria as it currently stands. Therefore, we invite you to submit a revised version of the manuscript that addresses the points raised during the review process.

I could find only one reviewer to comment on your manuscript. In order to provide you with more information and some suggestions, I decided to take the effort of reviewing myself. Detailed suggestions can be found below. Reviewer 1 raised some methodological issues that might be subject of consideration during the revision process. I would invite you to prepare a revision of your work that addresses all concerns together with a cover letter that provides point-by-point replies. My own comments can be found below and are aimed to aid you in the process of revision.

We look forward to receiving your revised manuscript.

Kind regards,

Michael B. Steinborn, PhD

Academic Editor

PLOS ONE

**Additional Editor Comments:**

(-1-) Tests (--) More information on p. 8 (description of performance tests) is needed. For example, are trials in the PVT forewarned by a preparatory interval, termed foreperiod (see Langner et al., 2010)? If so, how long and variable is the interval? How long is the intertrial interval and how is it distributed (see Steinborn et al., 2016)? Could you specify the elementary trials events with a bit more precision? How is the target terminated, after response or time-based? (--) How are performance scores computed? What are the essential performance measures. Typically, a measure of RT mean is taken to index response speed while the reaction time coefficient of variation is typically used to index performance variability (cf. Flehmig et al., 2007), which seems to be of interest in the present study. Since it seems that the authors are not from the field of performance testing, I would suggest my own work that could potentially serve as a tutorial to guide you in the process of revision. I apologise for recommending my own work but it is frequently cited and might help you by providing a showcase of how to compute and interpret performance measures in chronometric tasks.   (-2-) Self-reports I agree with reviewer 1 that more information should be provided regarding how state was exactly measured in the present study. I would appreciate if the authors could provide more detail with this regard. I would like to leave some comments: It has increasingly become a standard procedure to obtain pretest and posttest assessments (before and after a performance measurement) of the fundamental dimensions of subjective "state" in performance settings such as the present situation. With this respect, I would like to suggest a psychometric instrument for assessing energetic state in performance settings, which has become the gold standard in many domains. The dundee stress state questionnaire (DSSQ, Matthews et al., 2002, see also Langner et al., 2010, for methodical aspects of assessing state using the psychomotor vigilance --simple-RT-- task as in the present study) is a theory-oriented instrument aimed to assess the fundamental dimensions of subjective state in performance settings, namely task engagement, distress, and worry. The measure is widely accepted and the instrument is well-evaluated. I would recommend the DSSQ for future studies, but more importantly, it would be appreciable if the authors could give a short outlook on the possibilities of assessing engagement to the task and to elaborate somewhat more deeply on potential limitations with this regard in the present study.       (-3-) Variability  Reviewer 1 suggests specifying the use of variability analysis (within and between participants) in the present manuscript. Regarding the assessment of variability due to lapsing in simple reaction time tests, more information is needed. To measure variability, it is conventional to use an interpolated cumulative distributive function (CDF), sometimes referred to as percentile-point function, instead of defining a fixed criterion. Undoubtedly given that these functions are relatively sophisticated (thus require a certain level of experience with reaction-time based methods), I do not demand CDFs in the revised version of the manuscript. However, I would urgently advocate considering these methodical issues in the discussion of the revised manuscript. A viable alternative to percentile-point functions is to adopt a relativized measure of reaction-time variability (e.g., the coefficient of variation, RTCV), to indicate lapsing (in terms of performance instability). My work might serve as methodological guide at this point (Steinborn et al., 2016).  (-4-) References  Steinborn, M. B. et al. (2018). Methodology of performance scoring in the d2  sustained-attention test: Cumulative-reliability functions and practical guidelines. Psychological Assessment,  30(3), 339-357. doi:10.1037/pas0000482  Langner, R. (2010). Mental fatigue and temporal preparation  in simple reaction-time performance. Acta Psychologica, 133(1), 64-72. doi:10.1016/j.actpsy.2009.10.001  Steinborn, M. B. et al. (2016). Everyday life cognitive instability predicts simple  reaction time variability: Analysis of reaction time distributions and delta plots. Applied Cognitive Psychology,  30(1), 92-102. doi:10.1002/acp.3172  

2. Peer review at PLOS ONE is not double-blinded (https://journals.plos.org/plosone/s/editorial-and-peer-review-process). For this reason, authors should include in the revised manuscript all the information removed for blind review.

3. Please consider changing the title so as to meet our title format requirement (https://journals.plos.org/plosone/s/submission-guidelines). In particular, the title should be "Specific, descriptive, concise, and comprehensible to readers outside the field" and in this case the acronym CCT may not be comprehensible to readers outside the field.

Reviewers' comments:

Reviewer's Responses to Questions

**Comments to the Author**

1. Is the manuscript technically sound, and do the data support the conclusions?

Reviewer #1: Partly

2. Has the statistical analysis been performed appropriately and rigorously? 

Reviewer #1: No

3. Have the authors made all data underlying the findings in their manuscript fully available?

Reviewer #1: No

4. Is the manuscript presented in an intelligible fashion and written in standard English?

Reviewer #1: Yes

5. Review Comments to the Author

Reviewer #1: The writing is good and the the manuscript is well organized. I cannot judge whether the research question is important as I am not the particular expert in the field of architectural psychology. I can judge the overall design and the use of performance tests, which covers my expertise. My concerns should be considered before publication.

## the writing is good, however, the theoretical point, lack of knowledge in the field of research, or general aims of the study are held to vague and should be specified. what is the rationale of the study? What is aimed to find out?

## The hypotheses or research questions should be specified in more detail?

## The design is not clear to me. I suggest providing a figure that displays the design features.

## More information should be provided regarding the used tests and performance measures, and also, the self-report instruments used to study subjective feeling. I see that the authors used the PVT which is basically a simple reaction time task to study elementary functions. More literature should be provided and the tasks used should be described in more detail.

## Statistics should be presented in tables and in a more systematic way. Measurements are taken on more than one occasion to study consistency or reliability of performance and subjective ratings? It is asked whether there is variation between the sample population, it is not clear what is meant here? There is always variability between the participants, is it aimed to ask whether this variation increases? Please specify.

## There are too many Figures. I suggest using larger Figures with panel, to enable the comparison of important information at a glance. The scaling should be the same in different figures, otherwise, how can one compare the information provided in the figures.

## The discussion is a good read. Yet, the interpretation should be reconsidered a bit.

6. PLOS authors have the option to publish the peer review history of their article (what does this mean?). If published, this will include your full peer review and any attached files.

Reviewer #1: No

---

## [Author Response · Author response to Decision Letter 0]

5 Mar 2021

Response to the editors comments:

 (-1-) Tests

Thank you for these comments and suggestions.

The performance score for the BDST task was computed by summing each correct response. For the PVT task we calculated the mean of the response times, after removing outliers in the respective trial (responses that deviated more than 3 SD from the mean in the respective trial). We have extended the section explaining the tasks in the manuscript as shown in the text below.

Thank you for pointing us to the relevance of RT variability metrics. In response to this, we have taken steps to examine this in our data and elaborate on this below (under (-3- Variability)).

(-2-) Self-reports

The goal of this study was to examine the temporal trajectories of various variables before and during transitions in light settings. These measures (including subjective reports of visual and thermal experience, sleepiness, vitality and mood, but also the performance on the PVT and BDST, and physiological measurements) were repeatedly probed throughout the experiment to gain insight in the development of these variables over time throughout the experiment. Our aim was not to additionally measure how performance tasks can change subjective states, therefore we did not measure these states immediately before and after the performance tasks.

However, subjective states may indeed also be affected by performing tasks and this is exactly the reason why we first asked participants to complete the self-reports and then they had to perform the tasks. After the two tasks participant had a few minutes to recover before the next block with self-reports and performance tasks started. As the tasks lasted only 9 minutes (5min PVT, 4min BDST), we expect that the effect of performing the task on the subjective states is relatively small. Still, participants’ subjective state may have been influenced by doing the performance tasks and this effect would then in the current study be confounded with the effect of block (or the time within the session).

Furthermore, motivation and effort also influence task performance. We therefore also assessed the exerted effort on the task with visual analogue scales (VAS) ranging from None (0) to Very Much (20). As the indicated effort correlated highly with the effort that was done for the reading in between tasks, we added only this ‘Reading Effort’ in the analyses. We now have additionally run the analysis for the effort for the PVT and BDST and added these in the results section. For both the PVT and the BDST the exerted effort was indeed lower in the bright light condition, albeit not only significantly so. These results suggest that the light effect may have been compensated (at least partially) for by a drop in effort. Additionally, we embedded a baseline measurement in each session to account for potential differences in motivation and effort between persons and within persons between sessions. Rather than calculating difference scores, we chose to include this baseline assessment in the models to account for this variation. We were not aware of the existing questionnaires measuring task engagement, thus we would like to thank you for your suggestion and definitely take the proposed assessment of task engagement, distress and worry into consideration for future studies.

In the discussion we have elaborated on these aspects.

(-3-) Variability

The variability of the responses within the 5min of the PVT is indeed currently not taken into account. Some other papers in the field have used the 10% fastest and slowest responses. However, as in the 5 minutes PVT only 15-20 trials could be completed 10% would refer to 1 or 2 responses only, which we regarded too little for an insightful measure. Lapses were also counted but since the number of lapses was very low in general (see Table 1) and hence showed very limited variance, we did not include these as an outcome parameter. We have now also calculated the coefficient of variation of the reaction time (CVRT) and the standard deviation of the reaction time (SDRT) and investigated the distribution of these variables. Both had high kurtosis values (6.8 for SDRT and 5.9 for CVRT) demonstrating the lack of variation in these parameters. This can presumably be explained by the duration of the task (5 minutes) and the relatively large ISI, which together resulted in a relatively low number (15-20) of trials per task. 

For these reasons, we focused on only the mean reaction time in ms as output parameter. We thank the editor for informing us about these measures and will examine the suitability of these parameters in subsequent research. We also reflect on this in the limitations section of the discussion.

Reviewers comments:

## the writing is good, however, the theoretical point, lack of knowledge in the field of research, or general aims of the study are held to vague and should be specified. what is the rationale of the study? What is aimed to find out?

## The hypotheses or research questions should be specified in more detail?

Thank you for making these comments. We realized that for the general audience that PloS One has we need to provide more background to our introduction. Therefore, we have extended the introduction with a short introduction on the effects of light, and tried to be more explicit throughout the entire introduction in explaining our aims.

Illuminance and CCT are two important characteristics describing the light in an environment and these two are often manipulated in a dynamic light scenario. Dynamic light scenarios are upcoming both for commercial uses as well as in research, but studies targeting how the transitions in such dynamic light scenarios affect the user are lacking. This is the main motivation behind the current study. By using multiple measurements over a short time interval we gain insight in how the effects of transitions in intensity and/or CCT develop over time, as we believe that these subjective experiences, behavioral performance and physiological measures can vary considerably over time. We think it is important to not only decide at the end of a period how someone feels or behaves when reflecting on this period, but we believe it is important to measure these feelings and behaviors throughout the entire period. Therefore, we regard this as an important aspect to gain insight in these temporal trajectories. Additionally, we investigate to which extent these effects vary between participants to gain more insight in whether light scenarios could be similar for all, or should be tailored to the individual.

The research questions are:

To what extent does an abrupt transition from dim to bright light, or from warm to cool light, influence subjective evaluations of comfort and mood as well as subjective and objective measures of alertness, arousal and thermoregulation over time?

To what extent does systematic interindividual variability occur in these light-induced effects. 

To which extent do transitions in the illuminance and correlated colour temperature (CCT) of the light interact. 

We have formulated them more clearly in the introduction and added the main hypotheses.

## The design is not clear to me. I suggest providing a figure that displays the design features.

In total, we used four conditions (Fig 1: a) Cool, bright light; b) Cool, dim light; c) Warm, bright light; and d) Warm, dim light) to which all participants were exposed in separate sessions on separate days. The order of these conditions was counterbalanced. Each session lasts 90 minutes. The first 45 minutes of all sessions are identical (= the baseline period); in warm, dim light people are instructed, sensors are applied and participants practice the task. In the last 15 minutes of this baseline period participants complete one measurement block with self-reports, performance tasks and physiological measurements which is used as the baseline measurement. In the second 45 minutes of each session, the light is changed to one of the four conditions. In this experimental phase participant complete three more measurement blocks that are identical to the baseline measurement block.

The design of the study is visualized in Figure 1. 

## More information should be provided regarding the used tests and performance measures, and also, the self-report instruments used to study subjective feeling. I see that the authors used the PVT which is basically a simple reaction time task to study elementary functions. More literature should be provided and the tasks used should be described in more detail.

The tests and performance measures are now described in more detail. For the self-report instruments we also added the corresponding references in the text. 

## Statistics should be presented in tables and in a more systematic way. Measurements are taken on more than one occasion to study consistency or reliability of performance and subjective ratings? It is asked whether there is variation between the sample population, it is not clear what is meant here? There is always variability between the participants, is it aimed to ask whether this variation increases? Please specify.

We have added Table 3 with the statistics for the main predictors in the models for all dependent variables. Only the estimated marginal means/coefficients of the significant parameters are presented in text. In the supplementary materials, we present the other predictors, the full model statistics and the null model statistics for all variables. Additionally, contrast estimates for the main effects of Illuminance and CCT and the contrast estimates for Illuminance and CCT per measurement block (which are indicated in the figures too using * / + ) are reported for all variables in the supplementary materials. 

Measurements were taken on more than one occasion within each condition to study the change in the response to the light transition over time.

The variation between the participants that is mainly described in the current manuscript reflects the interindividual (between person) variability in responsiveness to the changes in light condition. Variability indeed always exists between participants, but we have seen in prior research that variability in preferences for light settings can be extreme. In this study we aimed to investigate whether the variability is of a magnitude that certain light settings/transitions are comfortable to some participants but uncomfortable to others. This could lead to insignificant overall effects, while specific subgroups of participants might benefit greatly from the light settings, where others might be hindered. We therefore hypothesized that marked interindividual variability would exist comfort, but also for alerting responses. Prior literature on the alerting effect of light during daytime has yielded inconsistent results and it is often suggested that this might be explained by such significant interindivdual variability (de Zeeuw et al., 2019; Smolders, Peeters, Vogels, & de Kort, 2018; Souman, Tinga, te Pas, van Ee, & Vlaskamp, 2018). Alerting effects that certain light settings may have could vary over an order of magnitude between individuals (Phillips et al., 2019). The question we aimed to explore indeed was not whether there was interindividual variability, but to which extent it was present in the data and whether it could explain the null effects that have been reported in literature (Lok, Smolders, Beersma, & de Kort, 2018; Souman et al., 2018).

We have added some additional details regarding the rationale for the repeated measurements and the interindividual variability in responsiveness to the light manipulations in the introduction of the manuscript.

## There are too many Figures. I suggest using larger Figures with panel, to enable the comparison of important information at a glance. The scaling should be the same in different figures, otherwise, how can one compare the information provided in the figures.

We combined related figures into panels. We are happy to combine more figures into a larger panel but would like to ask the editor for his opinion on the size of the panel that would be most suitable in the final lay-out of the article.

As not all variables have the same scale, the scaling cannot be the same in all figures. We have adjusted the scaling in such a way that the figures that do describe the same variable have the same scale.

## The discussion is a good read. Yet, the interpretation should be reconsidered a bit.

With all your comments kept in mind we have taken another critical look at our discussion and revised the text in an attempt to present the interpretation of our results more clearly.

---

## [Editor Report · Decision Letter 1]

9 Mar 2021

Abrupt light transitions in illuminance and correlated colour temperature result in different temporal dynamics and interindividual variability for sensation, comfort and alertness

PONE-D-20-36022R1

Dear Dr. Kompier,

We’re pleased to inform you that your manuscript has been judged scientifically suitable for publication and will be formally accepted for publication once it meets all outstanding technical requirements.

Kind regards,

Michael B. Steinborn, PhD

Academic Editor

PLOS ONE

Editorial comment: (--) check for typos, examples, p. 4, line 91, applies here and on other occasions, please check: use e.g. only in brackets (e.g., ....), while in the normal text, use "for example,  p. 10, line 224, "...without foreperiod", in the PVT, individuals use the variable interstimulus intervals to separate trials from each other, and individuals are assumed to perceive them as "foreperiod", as evidenced by recent studies (suggested citations on this occasions: doi:10.1002/acp.3172, doi:10.1037/xlm0000712).  p. 13, table 3, no overlap of tables across pages p. 11-14, statistical indicators in italics (e.g., F, p, M, SD, etc.) p. 26, references  line 725, ref40 - doi numberline 790, re62 - doi number line 798, ref65 - authors, years 2018, doi:10.1037/pas0000482

---

## [Editor Report · Acceptance letter]

12 Mar 2021

PONE-D-20-36022R1 

Abrupt light transitions in illuminance and correlated colour temperature result in different temporal dynamics and interindividual variability for sensation, comfort and alertness 

Dear Dr. Kompier:

I'm pleased to inform you that your manuscript has been deemed suitable for publication in PLOS ONE. Congratulations! Your manuscript is now with our production department. 

Kind regards, 

on behalf of

Dr. Michael B. Steinborn 

Academic Editor

PLOS ONE